# Synthetic Causal Priors for In-Context Time-Series Classification

**Hao-Run Cai** [1 2]    **Han-Jia Ye** [1 2]

## Abstract

Time-series classification (TSC) tasks differ in sensors, sequence lengths, channel counts, and label semantics, making support-conditioned in-context prediction a natural interface. We study how to generate synthetic classification episodes for this setting. Our generator unifies Kernel-Synth temporal templates, SCM mechanisms, label rules, support/query sampling, and optional multichannel observation. The main empirical lesson is *label visibility*: a label can be causally meaningful, but transfer depends on whether its effect is recoverable from the observed series. On the binary UCR-42 subset, an enforced-path within-SCM label prior, where one SCM generates all classes through a hidden causal source, is the strongest full-support synthetic prior in our comparison: it outperforms both a KernelSynth mixup prior and a class-SCM prior, without using target benchmark series during pretraining.

## 1. Introduction

Time-series classification (TSC) underpins applications in medical monitoring, industrial sensing, human motion analysis, and environmental observation, where models assign labels to temporal patterns collected from heterogeneous sensors and domains (Bagnall et al., 2017; Fawaz et al., 2019; Dau et al., 2019). These tasks are heterogeneous: datasets differ in sampling rate, sequence length, channel count, sensor semantics, class semantics, noise level, and label budget. A model trained for one dataset often cannot be reused on another without adapting the task interface. This makes TSC a natural target for in-context learning (ICL), where a model receives a labeled support set and predicts query labels in a single forward pass.

[1]School of Artificial Intelligence, Nanjing University, China. [2]National Key Laboratory for Novel Software Technology, Nanjing University, China. Correspondence to: Han-Jia Ye <yehj@lamda.nju.edu.cn>.

*Proceedings of the $2^{nd}$ ICML Workshop on Foundation Models for Structured Data*, Seoul, South Korea. 2026. Copyright 2026 by the author(s).

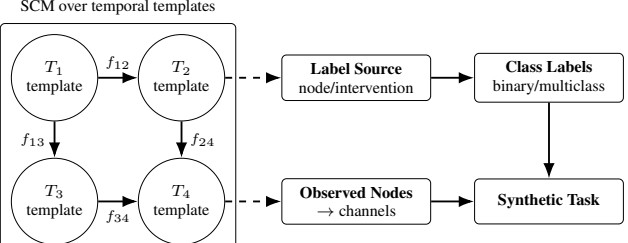

SCM over temporal templates

*Figure 1.* Synthetic causal task prior. Each node is a KernelSynth template, directed edges define SCM equations, labels come from a causal source or intervention, and descendant observations become channels in a support/query classification task.

Much of the recent progress in time-series foundation models has focused on forecasting, where future values provide natural supervision (Ansari et al., 2024; Das et al., 2024). Synthetic data have mostly entered this line through prediction: Gaussian-process kernel mixtures and related generators produce diverse trajectories for next-value or reconstruction objectives (Liu et al., 2025). Classification is less direct. Labels do not arise automatically from a trajectory; a synthetic TSC prior must also specify class semantics, support/query splits, class balance, and task-level mechanisms. This introduces a central design variable we call *label visibility*: the label source should be meaningful, but its effect must remain recoverable from the observed time series.

Prior-data-fitted networks (PFNs), especially TabPFN, suggest a route: train a transformer on synthetic classification tasks so that inference becomes conditional prediction from a support set (Hollmann et al., 2023; 2025; Qu et al., 2025). Recent TSC systems address different parts of this problem. Mantis- and MOMENT-style pipelines learn frozen encoders (Goswami et al., 2024; Feofanov et al., 2025) and still fit a task-specific classifier. TimEE trains an end-to-end ICL classifier, but constructs pretraining tasks from real UCR training splits (Küken et al., 2026). TiCT trains on KernelSynth mixup tasks (Yeh et al., 2025). CauKer combines kernel composition with structural causal models (SCMs) for synthetic classification pretraining (Xie et al., 2026). These works motivate a narrower question: *what synthetic task prior should produce classification episodes for time series, and what properties make such episodes transfer?*

We study this question by isolating the **synthetic task prior** under a fixed in-context classification interface, together

with a **diagnosis of evaluation protocols**. The prior combines KernelSynth temporal templates, DAG/SCM mechanisms, and label rules that define classes from template mixing, class-specific SCMs, hidden causal variables, or class-dependent interventions. We compare three aligned prior families: a *KernelSynth mixup prior*, where labels are induced directly from template mixtures; a *class-SCM prior*, where each class selects a different SCM mechanism; and a *within-SCM label prior*, where a single SCM generates all classes through a hidden label source or intervention.

**Contributions.** We make three contributions toward understanding synthetic task priors for in-context TSC. (1) We specify a unified synthetic causal task prior that connects temporal templates, SCMs, label generation, support/query sampling, and multichannel observation (Figure 1). (2) We identify label visibility as a practical explanation for why within-generator learnability and real-data transfer can diverge: labels must be generated by meaningful mechanisms and remain observable after SCM propagation. (3) We provide a protocol diagnosis showing that retrieval and dataset splitting are part of the evaluation interface and must be reported explicitly. Empirically, the enforced-path within-SCM label prior is the strongest full-support synthetic prior in our comparison, outperforming the KernelSynth mixup and class-SCM priors on the binary UCR-42 subset while remaining within $4.7$ accuracy points of an in-domain real-data mixup reference that uses target benchmark training distributions.

## 2. Background and Related Work

**Prior-data-fitted models.** ICL was popularized by language models and later studied in supervised settings where transformers learn predictors from examples in context (Brown et al., 2020; Garg et al., 2022). TabPFN turns this idea into a supervised learning system by fitting a transformer to synthetic tabular tasks (Hollmann et al., 2023; 2025); recent work scales tabular priors or adapts PFN-style training to forecasting (Liu & Ye, 2025; Dooley et al., 2023; Taga et al., 2025b; Hoo et al., 2025). We use the same prior-data-fitted view, but the prior must generate classification episodes: trajectories, labels, class semantics, and support/query structure.

**Synthetic time-series data.** Synthetic time series are widely used for pretraining, augmentation, and controlled probes (Liu et al., 2025). Forecasting systems such as Chronos, TimesFM, MOMENT, UniTS, and Moirai use synthetic, mixed synthetic-real, or large heterogeneous corpora to cover temporal patterns (Ansari et al., 2024; Das et al., 2024; Goswami et al., 2024; Gao et al., 2024; Woo et al., 2024). Synthetic-only ForecastPFN and TimePFN further show that prior-data fitting can work without large real pretraining corpora (Dooley et al., 2023; Taga et al., 2025a).

These systems mostly learn from values to predict values. In TSC, the prior must additionally decide what the classes mean and how labels affect observations.

**Time-series classification priors.** The UCR and UEA archives remain standard TSC testbeds (Dau et al., 2019; Bagnall et al., 2018). Classical and neural systems include shapelets, ROCKET, InceptionTime, and HIVE-COTE (Ye & Keogh, 2009; Bagnall et al., 2017; Dempster et al., 2020; Fawaz et al., 2019; 2020; Lines et al., 2018). Recent foundation-style TSC methods use different interfaces: frozen encoders with task-specific classifiers (Goswami et al., 2024; Feofanov et al., 2025; Auer et al., 2025), ICL heads on pretrained representations (Fang et al., 2026), and end-to-end ICL systems such as TimEE and TiCT (Küken et al., 2026; Yeh et al., 2025). TimEE supplies an ICL interface but builds tasks from real UCR training data. TiCT is synthetic but uses KernelSynth mixup task construction. CauKer supplies kernel and SCM ingredients; in our taxonomy it corresponds most closely to a class-SCM prior, where classes select different SCM mechanisms (Xie et al., 2026). Our focus is the missing glue: a within-SCM label prior that links temporal templates, SCM edges, label rules, support/query sampling, and multichannel observation while generating all classes from one task-level SCM.

## 3. Problem Setup

A time-series classification task is a pair of support and query sets,

$$\mathcal{S} = \{(x_i, y_i)\}_{i=1}^{n_s}, \qquad \mathcal{Q} = \{x_j\}_{j=1}^{n_q}, \qquad (1)$$

where each $x \in \mathbb{R}^{C \times T}$ may be univariate or multivariate. An ICL classifier parameterized by $\theta$ estimates

$$p_\theta(y_j \mid x_j, \mathcal{S}) \qquad (2)$$

without updating parameters on the target task. In prior-data-fitted training, tasks are sampled from a prior $p(\tau)$ rather than from target benchmark datasets. The training loss is the average query cross-entropy over synthetic tasks:

$$\mathcal{L}(\theta) = \mathbb{E}_{\tau \sim p(\tau)} \left[ -\frac{1}{|\mathcal{Q}|} \sum_{(x,y) \in \mathcal{Q}} \log p_\theta(y \mid x, \mathcal{S}) \right]. \quad (3)$$

At evaluation time we use *full-support ICL*: all labeled examples in the official training split are placed in $\mathcal{S}$, and each query is predicted conditionally on the same support set. This differs from fitting a task-specific classifier head, because task information enters through context rather than parameter updates. The central design problem is therefore the synthetic task prior $p(\tau)$: it must define temporal variation, class-generating mechanisms, label semantics, and support/query sampling.

*Table 1.* Synthetic task-prior specification. Defaults for reported synthetic-prior runs.

| Component | Specification |
|---|---|
| Graph nodes | $|V| \in [C+1, 18]$; random topological order |
| Edge prior | Each node samples up to 6 earlier parents uniformly |
| Kernels | 1–7 kernels from trend, smooth, periodic, and noise banks |
| SCM maps | Aggregation in {sum, product, min, max}; nonlinearities include linear, ReLU, sigmoid, sin, mod, and lag |
| Lag | Integer delay in $[1, 16]$ when lagged dynamics are selected |
| Classes | 50% binary; otherwise $K \in \{3, \dots, 10\}$ uniformly |
| Support/query | Fixed per run; reported runs use $(192, 64)$ |
| Channels | $C \geq 1$ observed descendants; UCR experiments use $C = 1$ |
| Intervention | Class-dependent node states or mechanism parameters; optional enforced label–observation path |

---

**Algorithm 1** Sampling a synthetic causal ICL task

---

1: Sample $K$, graph $G = (V, E)$, templates $\{z_v(t)\}_{v \in V}$, and SCM equations $\{f_e\}_{e \in E}$.
2: Select prior family $p$, label source $\ell$ when applicable, and observed descendants $O = \{o_1, \dots, o_C\}$.
3: **for** $i = 1, \dots, n_s + n_q$ **do**
4:    Sample class $y_i$ and intervention/regime variables according to $p$.
5:    Propagate the class-dependent mechanism through $G$.
6:    Emit $x_{i,c}(t) = z_{o_c}^{(i)}(t)$ for each observed channel $o_c \in O$.
7: **end for**
8: Return a support/query split of $\{(x_i, y_i)\}$.

---

## 4. Synthetic Causal Priors

Our prior generator has three levels (Figure 1). First, **temporal templates** initialize latent nodes from kernel compositions and simple mean functions, preserving the Kernel-Synth intuition that Gaussian-process kernels provide controllable trend, periodicity, and smoothness. Second, **SCM mechanisms** sample directed edges between templates; parent templates are aggregated through structural equations and activations, optionally including lagged effects. Third, **label generation** selects a template-mix coefficient, a class-specific SCM, a hidden causal node, or a class-dependent intervention as the class source. Observed series are descendant nodes read out as one or more channels. This construction lets one latent causal task emit both univariate and multichannel classification episodes, although our reported UCR experiments are univariate.

The important design choice is that labels are not appended as a separate feature. A class can change a kernel mixture, a node state, an edge parameter, or a lagged response, and the classifier must infer the label from the emitted descendants. For multichannel tasks, observed channels are multiple descendants of the same latent graph, so channels may share causes, inherit lagged effects, or respond differently to the same class intervention.

**Prior-family definitions.** Let $G = (V, E)$ be the sampled DAG, $O \subset V$ the observed nodes, and $x_i = \{z_o^{(i)}(t) : o \in O\}$ the emitted series. *KernelSynth mixup* samples two templates $a, b$, a mixing weight $\lambda_i$, and emits $x_i = \lambda_i a + (1 - \lambda_i) b$ with $y_i = \mathbf{1}[\lambda_i > 1/2]$ or its multiclass binning variant. *Class-SCM* samples $y_i = k$ and uses a class-specific SCM mechanism, written as a class-specific graph or parameter set $(G_k, \phi_k)$, so the label is the identity of the SCM that generated $x_i$. *Within-SCM label* samples one graph $G$ for the whole task, samples a latent label node $\ell \in V$ and value $y_i = k$, intervenes on $\ell$ with a class-dependent state, and emits descendants $O \subset \text{Desc}(\ell)$; the label is causal but only visible through descendants. The enforced-path variant explicitly selects at least one observed descendant of $\ell$, preventing labels from being hidden in parts of the graph that never reach the emitted series. These families differ in where class identity enters the generator: template mixing, class-specific SCMs, or labels inside a shared SCM. The downstream ICL format and model input remain identical.

**Task construction.** Each prior instance generates a support/query task with fixed budgets. For binary and multiclass training, we use a mixed class-count prior: half of the tasks are binary, and the rest sample 3–10 classes. This keeps emphasis on common binary benchmarks while exposing attention-based classifiers to arbitrary class counts. At evaluation time, the default interface is full-support ICL: the entire training split is provided as support, and queries are predicted without task-specific parameter updates.

## 5. Experiments

**Benchmarks and protocols.** We evaluate on the UCR Archive, using the official train/test splits. The main prior comparison is reported on the binary UCR-42 subset used by TimEE; protocol diagnostics also use the full UCR archive (128 datasets). Unless otherwise stated, the protocol is *full-support ICL*: the official training split is the support set and the official test split is the query set. We report mean accuracy and ROC-AUC across datasets. Retrieval-based inference is reported separately as an evaluation-interface diagnostic.

**Model and training setup.** All prior comparisons use the same ICL classifier architecture, preprocessing, training budget, and evaluation code. Episode budgets follow the reported prior settings in Table 1; the intended experimental variable is the sampled task prior: KernelSynth mixup prior, class-SCM prior, enforced-path within-SCM label prior, or unconstrained within-SCM label prior. This keeps differences in Table 2 attributable primarily to the prior rather than to downstream fitting. We do not train a dataset-specific head on the target benchmark. At test time, queries are batched for memory efficiency, but every query batch conditions on the same official support split under full-support evaluation.

**Label visibility.** We use label visibility as the main anal-

*Table 2.* Binary UCR-42 results under the official split. Real-data mixup is an in-domain reference; synthetic priors use no benchmark series during pretraining.

| Pretraining source | Inference | Acc. | ROC-AUC |
|---|---|---|---|
| *In-domain reference* | | | |
| Real UCR mixup reference | full-support | 0.823 | 0.852 |
| *KernelSynth and class-SCM priors* | | | |
| KernelSynth mixup prior | full-support | 0.760 | 0.799 |
| KernelSynth mixup prior | retrieval | 0.788 | 0.828 |
| Class-SCM prior | full-support | 0.715 | 0.746 |
| *Within-SCM label priors* | | | |
| Within-SCM label, enforced path | full-support | 0.776 | 0.815 |
| Within-SCM label, enforced path | retrieval | 0.802 | 0.846 |
| Within-SCM label, unconstrained | full-support | 0.708 | 0.742 |

*Table 3.* Protocol diagnosis for the TiCT evaluation protocol.

| Protocol | Full UCR Acc. | binary UCR-42 Acc. |
|---|---|---|
| TiCT re-split + retrieval | 0.789 | 0.886 |
| Official split + retrieval | 0.727 | 0.859 |
| Official split + full-support | 0.683 | 0.837 |

ysis concept. A task has high label visibility when the observed series contain a stable, recoverable signal of the sampled class after nuisance variation, SCM propagation, and channel selection. Class-SCM tasks usually have high visibility because each class directly changes the generator. Within-SCM label tasks are more causally meaningful but can have lower visibility: the class may affect observations only through weak, nonlinear, or confounded descendants. KernelSynth mixup tasks have direct label visibility through the mixing coefficient, but lack causal class semantics.

This notion separates three cases that are easy to conflate. A prior can be learnable inside the generator because the label is strongly exposed; it can be causal but not learnable because the class source is hidden from observations; or it can transfer better because the visible signal resembles how real labels affect observed series. The experiments below should be read as a diagnosis of these cases rather than as a single leaderboard.

**Synthetic prior comparison.** Table 2 shows that synthetic priors learn measurable ICL behavior. The strongest full-support synthetic result is the enforced-path within-SCM label prior at 0.776 accuracy and 0.815 ROC-AUC, improving over the KernelSynth mixup prior (0.760, 0.799) and the class-SCM prior (0.715, 0.746). The in-domain real-data mixup reference remains higher at 0.823 and 0.852, but it uses target benchmark training distributions; the synthetic priors transfer from generated tasks only. The class-SCM prior has high within-generator visibility because each class directly selects an SCM, but this shortcut can transfer poorly when real datasets do not expose the same class-specific mechanism. The enforced-path within-SCM prior is harder than the class-SCM prior, yet transfers better because the label acts through observed causal descendants. The unconstrained within-SCM prior shows the opposite failure mode: causal labels without sufficient observed visibility are difficult to recover.

**Evaluation protocol matters.** Table 3 isolates the effect of evaluation protocol. Moving from TiCT's re-split to the official UCR split reduces mean accuracy on the full UCR archive by 6.2 points under retrieval. Removing retrieval further changes performance. These are not invalid choices, but they are part of the evaluation interface: retrieval changes which support examples each query sees, and re-splitting changes the target distribution. We therefore use the full-support official split by default, while reporting retrieval separately.

For an ICL classifier, the support set is an input to the model rather than only a storage format for labels. Full-support evaluation asks whether the pretrained model can condition on all labeled examples provided by the official task. Retrieval adds a query-dependent support-selection rule before prediction: each query is compared to the training split in the raw adapted series space, the nearest labeled examples are selected, and the model then predicts from that smaller support context. Reporting these settings separately keeps the prior comparison from being mixed with an additional test-time interface choice.

**Limitations.** The experiments isolate the task-prior and protocol questions. The prior supports multichannel observation by construction, but the reported UCR experiments are univariate. A multichannel follow-up should keep the same support/query protocol while varying channel graphs, shared confounders, and class-dependent cross-channel effects. Scaling to genuinely multichannel benchmarks and more efficient offline generation remains future work.

# 6. Conclusion

We presented a unified synthetic task prior for in-context time-series classification. By combining KernelSynth temporal templates, SCM mechanisms, label rules, support/query construction, and multichannel observation, the prior makes explicit where class information enters a synthetic task and how it reaches the observed series. The central finding is label visibility: causal structure alone is insufficient unless the label effect is observable after SCM propagation. This perspective explains the gap between class-SCM shortcuts, unconstrained within-SCM labels, and enforced-path within-SCM labels. Our protocol diagnosis also shows that retrieval and re-splitting materially change the conditioning interface. Future work should scale controllable synthetic priors and test them on genuinely multi-channel benchmarks.

## Acknowledgements

This work is partially supported by National Key R&D Program of China (2024YFE0202800).

## Impact Statement

Synthetic pretraining can reduce reliance on proprietary or privacy-sensitive time-series corpora. However, synthetic priors may encode unrealistic assumptions, so downstream use should report benchmark protocols and validate transfer on real data.

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
