# OpenReview forum: "Synthetic Causal Priors for In-Context Time-Series Classification"
_ICML.cc/2026/Workshop/FMSD — FMSD @ ICML 2026 Poster_

### Official Review · Reviewer_jGVw · 2026-05-21
**Interesting Evaluation of Synthetic Causal Priors for In-Context Time-Series Classification with a Focus on Label Visibility and Evaluation Protocols**

**Rating:** 8
**Confidence:** 4

**Review:**

## **Summary of contributions**

The paper studies Synthetic Causal Priors for In-Context Time-Series Classification.
It argues that generating meaningful TSC datasets is not trivial because labels are not directly tied to trajectories. The authors propose a unified causal framework where labels are generated from latent mechanisms but remain recoverable from observed signals, emphasizing a key property called label visibility. They evaluate how different synthetic priors, varying in label visibility, affect the performance of a fixed ICL model. This provides insights into how to design effective priors. Furthermore this paper analyzes the impact of evaluation protocols. Experiments are conducted on UCR-128 and a 42-task binary UCR subset.
The experiments highlight two main constraints for synthetic TSC pretraining: labels should arise from causal mechanisms while remaining identifiable from observations.
Evaluation choices (e.g., retrieval and dataset re-splitting) must be clearly specified since they can substantially affect reported results.


## **Strengths and weaknesses**

### **Strengths**
- Interesting proposal of a new prior for TSC.
- Conceptual insight on label visibility as a driver of generalization.
## **Suggestions**
- Provide more details on split strategies and on related works (especially TiCT).

---

### Official Review · Reviewer_j1c7 · 2026-05-21
**Analyzing Synthetic Causal Priors for Time Series Classification**

**Rating:** 5
**Confidence:** 3

**Review:**

The paper introduces a framework to evaluate which design decisions in the creation of SCMs for synthetic data generation matter most for downstream time series classification tasks. The paper highlights two important findings: labels must remain inferable from the latent structure, and evaluation protocol choices, particularly dataset splits, can substantially affect reported performance. These findings provide useful insight into how synthetic data generation pipelines should be designed to maximize downstream classification performance and improve evaluation reliability.
Nonetheless, the overall contribution feels somewhat limited and could be strengthened with additional ablations and analysis. While the paper shows that labels should be inferable from the latent space, it remains unclear how strong these dependencies need to be. For example, it would be interesting to analyze the effect of the number of confounders, causal hops, or edge strengths on downstream performance. Similarly, a broader study of which SCM design decisions matter most would make the conclusions more actionable. A dataset level breakdown would also strengthen the paper, particularly identifying which datasets are most sensitive to the synthetic generation process and whether this correlates with specific properties of the underlying SCM.

---

### Official Review · Reviewer_SGjQ · 2026-05-22

**Rating:** 6
**Confidence:** 2

**Review:**

The paper uses existing synthetic data creation methods to pretrain a time series classification model. The paper attempts to answer whether a synthetically pretrained time series classification model can generalize to real data.

The experiments are insightful. However, the paper does not clearly describe how the pretrained model predicts classes for real datasets, since the number of classes might vary across datasets.